# Effects of Nitrogen Form on Root Activity and Nitrogen Uptake Kinetics in *Camellia oleifera* Seedlings

Rui Wang [1,2], Zhilong He [1,2], Zhen Zhang [1,2], Ting Xv [1,2], Xiangnan Wang [1,2], Caixia Liu [1,2] and Yongzhong Chen [1,2,*]

1    Research Institute of Oil Tea Camellia, Hunan Academy of Forestry, Shaoshan South Road, No. 658, Changsha 410004, China
2    National Engineering Research Center for Oil Tea Camellia, Changsha 410004, China
*    Correspondence: chenyongzhong@hnlky.cn

**Abstract:** This study investigated the effects of nitrogen form on root activity and nitrogen uptake kinetics of *Camellia oleifera* Abel. seedlings, providing a scientific basis for improving nitrogen use efficiency and scientific fertilization in *C. oleifera* production. Taking one-year-old *C. oleifera* cultivar 'Xianglin 27' seedlings as subjects, 8 mmol·L$^{-1}$ of nitrogen in varied forms ($NO_3^-$:$NH_4^+$ = 0:0, 10:0, 7:3, 5:5, 3:7, 0:10) was applied in this study as the treatment conditions to investigate the effects of different nitrogen forms on root activity and nitrogen uptake kinetics in *C. oleifera* seedlings. Comparing the performance of nutrient solutions with different $NO_3^-$:$NH_4^+$ ratios, the results showed that a mixed nitrogen source improved the root activity of *C. oleifera* seedlings based on total absorption area, active absorption area, active absorption area ratio, specific surface area, and active specific surface area. When $NO_3^-$:$NH_4^+$ = 5:5, the total absorption area and active absorption area of the seedling roots reached the maximum. The results of uptake kinetic parameters showed that Vmax $NH_4^+$ > Vmax $NO_3^-$ and Km $NO_3^-$ > Km $NH_4^+$, indicating that the uptake potential of ammonium–nitrogen by *C. oleifera* seedlings is greater than that of nitrate–nitrogen. The conclusion was that compared to either ammonium– or nitrate–nitrogen, the mixed nitrogen source was better for promoting the root activity of *C. oleifera* seedlings, and the best nitrate/ammonium ratio was 5:5.

**Keywords:** *Camellia oleifera*; root activity; nitrate–nitrogen; ammonium–nitrogen; uptake kinetics



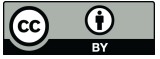

## 1. Introduction

*Camellia oleifera* is a woody edible oil tree species in China and mainly distributed in red soil regions which are characterized by soil depletion and nutrient deficiency. It is widely cultivated in south–central and southern China since their growing commercial, medic, cosmetic and ornamental values in recent years provide an important guarantee for a targeted poverty alleviation strategy in China [1]. The genome of oiltea-camellia is very complex and not well explored. Recently, genomes of three oiltea-camellia species were sequenced and assembled [2–4]; multi-omic [1,5,6] studies of oiltea-camellia were carried out and provided a better understanding of this important woody oil crop.

Nitrogen is an essential macronutrient for plant growth and development and has an irreplaceable role in plant life [7]. Nitrogen deficiency is an important factor limiting the growth and fructification of *C. oleifera* [8], which can be resolved by the scientific application of nitrogen fertilizers. Plants have a higher demand for nitrogen than for other elements, and the two main forms of nitrogen that can be absorbed and used by plants are ammonium–nitrogen ($NH_4^+$-N) and nitrate–nitrogen ($NO_3^-$-N). Although it is volatile, $NH_4^+$-N can be adsorbed and fixed by soil; thus, it is not easily lost via leaching. In comparison, $NO_3^-$-N is non-volatile but is prone to leaching and denitrification losses, which affects the use efficiency of nitrogen fertilizers. Selective uptake characteristics for different nitrogen forms and the kinetic characteristics of nitrogen uptake by roots are important factors affecting

nitrogen use efficiency in plants [9,10]. The kinetic approach is an effective method to study the nutrient uptake characteristics of plant roots and to identify the differences between different plant species [11]. In the early 1950s, Epstein et al. first applied the kinetic equation of enzymatic reactions to the study of nutrient ion uptake by plant roots [12], and in the mid-to-late 1970s, Barber et al. further modified the uptake kinetic equation and proposed the concept of critical concentration or minimum equilibrium concentration [13]. The Michaelis kinetic equation is a kinetic equation that can quantitatively characterize plant roots' nutrient uptake. This equation was used to investigate the effects of environmental conditions on nutrient uptake by plants [14]. The characteristic parameters obtained from this equation have important applications in comparing the barren resistance of different plant species (or varieties) [15,16]. To date, kinetic studies of nutrient ion uptake have been largely limited to crops and fruit trees [12,14,17,18]. There are few studies on nutrient ion uptake kinetics by roots of woody plants [19], and no relevant studies on *C. oleifera* have been reported.

In this study, different ratios of nitrate–/ammonium–nitrogen were applied to *C. oleifera* seedlings to investigate the root activity and root uptake kinetics under different nitrogen forms; these results would provide a theoretical basis for the fertilization of *C. oleifera* seedlings.

## 2. Materials and Methods

### 2.1. Overview of the Experimental Field

The experimental field is located at the experimental station of the National Oil-tea Camellia Engineering and Technology Research Center, at 113°01′ E, 28°06′ N, and 80–100 m above sea level. It belongs to a subtropical monsoon climate, with an annual average temperature of 16.8–17.3 °C, annual average rainfall of 1422 mm, a frost-free period of 275 days, and annual average relative humidity of 80%. The soil classifies as Quaternary red soil with a pH between 4.5 and 5.5, an organic matter content of 41.01 $g \cdot kg^{-1}$, a total nitrogen content of 2.68 $g \cdot kg^{-1}$, a total phosphorus content of 0.61 $g \cdot kg^{-1}$, and a total potassium content of 4.53 $g \cdot kg^{-1}$.

### 2.2. Experimental Materials

The *C. oleifera* cultivar 'Xianglin 27' one-year-old seedlings were used for the experiment. *C. oleifera* seeds were collected in October 2018, and uniform, full-grained, and disease-free seeds were selected for stratification. In March 2019, seeds with consistent germination states were selected and sowed in containers with a height of 12 cm and an upper diameter of 8 cm. The cultivation substrate consisted of yellow subsoil, perlite, and peat at a volume ratio of 3:1:1. The substrate had a pH of 5.88, and the concentrations of $NH_4^+$-N and $NO_3^-$-N were 0.92 and 2.34 $mg \cdot L^{-1}$, respectively. Slow seedling was performed for 3 months after planting seeds in nutrition cups, and seedlings with consistent growth conditions and no pests or diseases were selected for fertilization in June 2019.

### 2.3. Experimental Design

Fertilizers were applied via liquid irrigation, and a modified Hoalgland nutrient solution (nitrogen-free) was added to ensure that the seedlings grew under normal nutritional conditions. The formulation of the nutrient solution is: $K_2SO_4$ 261.39 $mg \cdot L^{-1}$, $KH_2PO_4$ 136.09 $mg \cdot L^{-1}$, $CaCl_2$ 221.98 $mg \cdot L^{-1}$, $MgSO_4 \cdot 7H_2O$ 246.47 $mg \cdot L^{-1}$, $MnSO_4 \cdot H_2O$ 1.54 $mg \cdot L^{-1}$, $H_3BO_3$ 2.86 $mg \cdot L^{-1}$, $ZnSO_4 \cdot 7H_2O$ 0.22 $mg \cdot L^{-1}$, $CuSO_4 \cdot 5H_2O$ 0.08 $mg \cdot L^{-1}$, $Na_2MoO_4 \cdot 2H_2O$ 0.02 $mg \cdot L^{-1}$, and $FeSO_4 \cdot 7 H_2O$ 20 $mg \cdot L^{-1}$. In all nutrient solutions, 7 $\mu mol \cdot L^{-1}$ of the nitrification inhibitor dicyandiamide ($C_2H_4N_4$) was added for nitrification inhibition.

The experiment employed a completely randomized block design. According to a previous study [20], six treatments (Table 1), including five experimental groups at a nitrogen level of 8.0 $mmol \cdot L^{-1}$ with different ratios of nitrogen forms ($[m(NO_3^-$-N$)/m(NH_4^+$-N$)]$ = 10:0, 7:3, 5:5, 3:7, 0:10), and a control group (no nitrogen fertilization) were set up.

For each treatment, three replicates, each containing 200 seedlings, were set up, as shown in Table 1. The first fertilization was carried out in mid-June, followed by 10 fertilizer applications at one-week intervals. Each seedling was thoroughly irrigated with 300 mL solution each time. The solution collected in trays was used for re-irrigation to ensure no nitrogen loss. The experiments were conducted in a greenhouse with a luminance of 6000–8000 lux, a temperature of 20.0–25.0 °C, and a humidity of 80%–85%. In addition to fertilization, other care and maintenance measures, such as normal watering and weed clearing, were performed.

**Table 1.** Nitrogen ratio under different treatments used in this study.

| No. | $m(NO_3^- \text{-} N)/m(NH_4^+\text{-}N)$ | Total Nitrogen/mmol·L$^{-1}$ | Nitrogen Sources/mmol·L$^{-1}$ | |
| --- | --- | --- | --- | --- |
| | | | NaNO$_3$ | (NH$_4$)$_2$SO$_4$ |
| A0(CK) | 0:0 | 0 | 0 | 0 |
| A1 | 10:0 | 8 | 8 | 0 |
| A2 | 7:3 | 8 | 5.6 | 1.2 |
| A3 | 5:5 | 8 | 4 | 2 |
| A4 | 3:7 | 8 | 2.4 | 2.8 |
| A5 | 0:10 | 8 | 0 | 4 |

*2.4. Experimental Methods*

The methylene blue method was used to determine root absorption activity [21]. The kinetic characteristics of the uptake of NO$_3^-$-N and NH$_4^+$-N solutions by seedlings were determined by the conventional depletion method from late July to the middle of August, which was the most active stage of seedlings' growth. The content of NO$_3^-$-N was determined using the colorimetric method [22], while that of NH$_4^+$-N was determined using the indophenol blue method [23]. Immediately after the uptake kinetics test, the *C. oleifera* seedlings were taken out and weighed after removing water from the root surface with absorbent paper.

*2.5. Experimental Materials*

The roots' kinetic uptake parameters were calculated using the method described by Hua Haixia and Zhai Mingpu. The maximum uptake rate $V_{max}$ and the Michaelis constant $K_m$ were calculated, and $\alpha = V_{max}/K_m$ [24,25]. Data were processed and statistically analyzed using Excel 2007 and SPSS25.0 software, and significant differences between treatments were compared using one-way analysis of variance (ANOVA) and the least significant difference (LSD) test ($p < 0.05$). Histograms were plotted with GraphPad Prism 8.0 and line graphs were plotted with OriginPro 8.5.1.

**3. Results**

*3.1. Effects of Nitrogen Form on the Root Activity of C. oleifera Seedlings*

3.1.1. Effect of Nitrogen Form on the Total Absorption Area of *C. oleifera* Seedling Roots

For simplicity, the nitrogen source treatments used in this study [$m(NO_3^-$-N)$/m(NH_4^+$-N)] = 10:0, 7:3, 5:5, 3:7, 0:10 will be referred to as A1–A5, respectively, and the control treatment will be referred to as A0. With the increasing proportion of NH$_4^+$-N, the total absorption area of the roots first increased and then decreased. This shows that the mixed-nitrogen-source treatment (A2–A4) had greater root absorption than the control (A0), which had a greater root absorption than the single-nitrogen-source treatment (A1, A5). For the single-nitrogen-source treatments, the total-nitrate treatment (A1) had a greater absorption than the total-ammonium treatment (A5). The total root absorption area of NO$_3^-$:NH+ 4 = 5:5, A3 treatment reached a maximum of 1.48 cm$^2$, which was 43.69% higher than the control, while the total absorption area of the total-ammonium (A5) treatment was the smallest at 0.85 cm$^2$, which was 17.48% lower than the control.

### 3.1.2. Effect of Nitrogen Form on the Active Absorption Area of *C. oleifera* Seedling Roots

As presented in Figure 1, the active absorption area of the roots also showed an increase, followed by a decreasing trend with increasing proportions of $NH_4^+$-N. Mixed-nitrogen-source treatment (A2–A4) had a greater active absorption area compared with the control (A0), which had a greater active absorption area than the single-nitrogen-source treatment (A1, A5). For the single-nitrogen-source treatments, A1 showed a greater active absorption area than A5. The active absorption area reached a maximum of 1.00 m$^2$ when using the A3 treatment, which was 78.57% higher than that of the control. The total-ammonium treatment resulted in the smallest active absorption area at 0.45 cm$^2$, which was 19.64% lower than the control. These results indicate that mixed nitrogen sources can significantly increase the active absorption area, and both total-nitrate and total-ammonium treatments will lower the active absorption area.

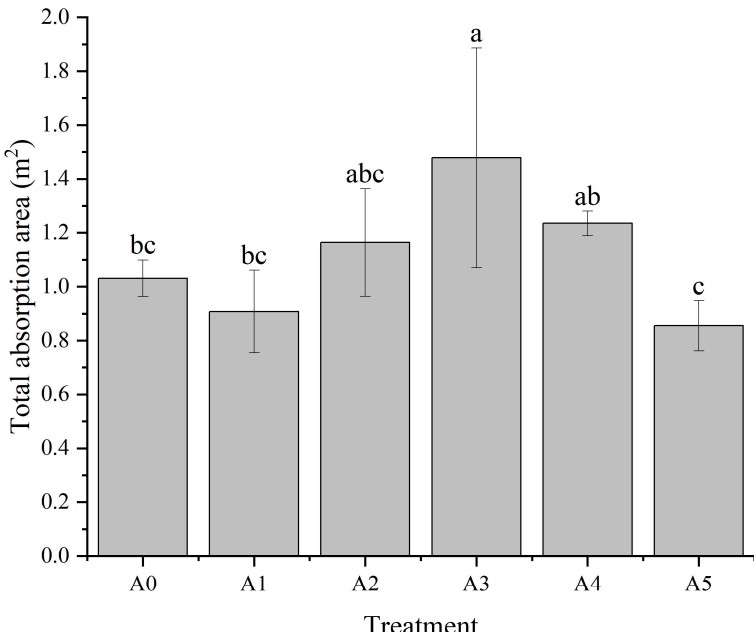

**Figure 1.** The effect of nitrogen form on the total absorption area of the roots of *C. oleifera* seedlings. Different lowercase letters indicate significant differences at $p < 0.05$.

### 3.1.3. Effect of Nitrogen Form on the Active Absorption Area Ratio of *C. oleifera* Seedling Roots

As shown in Figure 2, the active absorption area ratio tended to decrease, then increase, and then decrease with treatments A0–A5. Mixed-nitrogen-source treatments (A3, A4) had greater active absorption than the control (A0), which had greater active absorption than the single-nitrogen-source treatments (A1, A5). For the single-nitrogen-source treatments, A5 had greater active absorption than A1. The active absorption area ratio reached a maximum of 79.72% at $NO_3^-$:NH+ 4 = 3:7, which was 46.36% higher than that of the control, followed by that of A3 treatment with an active absorption area ratio that was 34.07% higher than the control's. There was no significant difference between A4 and A3 treatments, and both resulted in a significantly higher active absorption area ratio than the other treatments. The smallest active absorption area ratio, at 53.00%, was associated with the A5 treatment, which was 2.70% lower than that of the control. These results indicate that mixed nitrogen sources (A3 and A4) can significantly increase the active absorption area ratio.

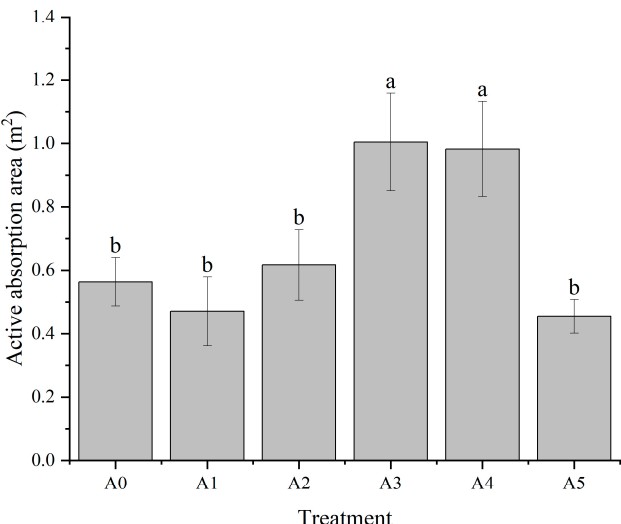

**Figure 2.** The effect of nitrogen forms on the active absorption area of the roots of *C. oleifera* seedlings. Different lowercase letters indicate significant differences at $p < 0.05$.

### 3.1.4. Effect of Nitrogen Form on the Specific Surface Area of *C. oleifera* Seedling Roots

The effect of nitrogen form on the specific surface area of *C. oleifera* seedling roots is shown in Figure 3. After treatments with nitrogen sources of different forms and ratios, the specific surface area of the roots showed a first increasing, then decreasing, and then increasing trend with an increasing proportion of $NH_4^+$-N. Specifically, seedlings had a greater root surface area after mixed-nitrogen-source treatment (A2, A4) compared with control (A0), which had a greater root surface area compared with seedlings after single-nitrogen-source treatment (A1, A5). Regarding the single-nitrogen-source treatments, A5 showed greater root surface area than A1. The root-specific surface area reached a maximum of 0.98 cm$^2 \cdot$cm$^{-3}$ in the A2 treatment, which was 30.67% higher than that of the control, followed closely by the A4 treatment. The root-specific surface area after the A2 treatment was significantly higher than those with single-nitrogen-source treatments (A1 and A5), but showed no significant differences compared to the other treatments. The A1 treatment led to the smallest root-specific surface area of 0.53 cm$^2 \cdot$cm$^{-3}$, which was 29.33% lower than that of the control. These results indicate that both total-nitrate and total-ammonium treatments will significantly reduce the specific surface area of *C. oleifera* seedlings.

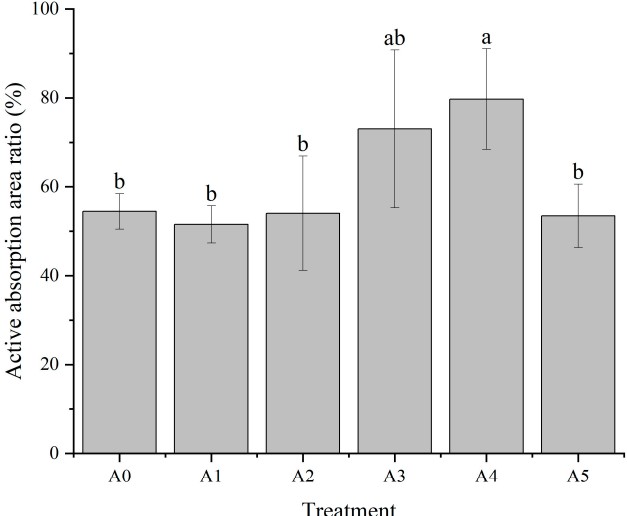

**Figure 3.** The effect of nitrogen forms on the active absorption area ratio of the roots of *C. oleifera* seedlings. Different lowercase letters indicate significant differences at $p < 0.05$.

### 3.1.5. Effect of Nitrogen Form on the Active Specific Surface Area of *C. oleifera* Seedling Roots

As shown in Figure 4, with the increased ratio of $NH_4^+$-N, the active specific surface area of the roots showed a first increasing and then decreasing trend. The active specific surface area of the roots reached a maximum value of 0.66 $cm^2 \cdot cm^{-3}$ under A4 treatment, which was 65.00% higher than the control. There was no significant difference between A4 and the other mixed-nitrogen-source treatments (A2 and A3), but the active specific surface area achieved by the A4 treatment was significantly higher than those by the single-nitrogen-source treatments (A1 and A5), and the control (A0). The active specific surface area was the smallest, at 0.27 $cm^2 \cdot cm^{-3}$, when A1 treatment was applied, which was 32.50% lower than the control. The results indicate that mixed nitrogen sources can significantly increase the proportion of active specific surface area, and both total-nitrate and total-ammonium treatments will reduce the active specific surface area of *C. oleifera* seedling roots.

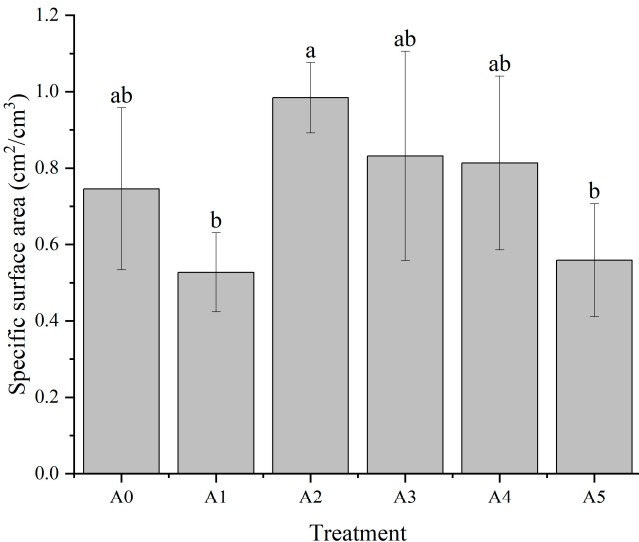

**Figure 4.** The effect of nitrogen forms on the specific surface area of roots of *C. oleifera* seedlings. Different lowercase letters indicate significant differences at $p < 0.05$.

### 3.2. Effects of Nitrogen Form on the Nitrogen Uptake Kinetics of C. oleifera Seedling Roots

3.2.1. Effect of Nitrogen Form on $NO_3^-$ Uptake Kinetics of *C. oleifera* Seedling Roots

Figure 5 shows the concentration curves of $NO_3^-$ at different uptake times in the $NO_3^-$ depletion test of *C. oleifera* seedlings treated with different nitrogen forms. The kinetic response patterns of $NO_3^-$-N uptake by seedlings subjected to different nitrogen treatments showed no significant difference. With the extension of the absorption time, the $NO_3^-$ concentration in the fertilizer solution gradually decreased. Specifically, the $NO_3^-$ concentration decreased rapidly within the first nine hours, with the most dramatic drop observed within the first two hours. From the ninth hour onwards, the $NO_3^-$ concentration remained almost constant. Via fitting, the $NO_3^-$ depletion equations for treatments with different nitrogen forms and ratios were listed in Table 2. The kinetic parameters of $NO_3^-$ uptake by the roots were calculated according to the fresh weights of the roots of *C. oleifera* seedlings in different treatments.

As shown in Table 2, the *Vmax* values of different treatments in descending order, are A3 > A1 > A4 > A2 > A0 > A5. The *Vmax* of the A3 treatment was the highest, at 17.57 $\mu mol \cdot g^{-1} \cdot h^{-1}$, indicating that the A3 treatment had the highest intrinsic potential for $NO_3^-$ uptake. The A1 treatment came in second place, with a *Vmax* of 17.52 $\mu mol \cdot g^{-1} \cdot h^{-1}$, followed by the other mixed-nitrogen-source treatments (A4 and A2). The A5 treatment resulted in the lowest *Vmax*, which was lower than that of the control. These results indicate

that the presence of $NH_4^+$ accelerates the uptake of $NO_3^-$; however, when $NH_4^+$ exceeds a certain percentage, the $NO_3^-$ uptake rate decreases.

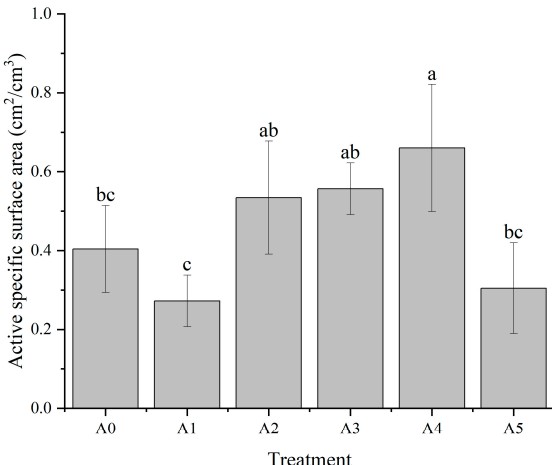

**Figure 5.** The effect of nitrogen forms on the active specific surface area of roots of *C. oleifera* seedlings. Different lowercase letters indicate significant differences at *p* < 0.05.

**Table 2.** Kinetic parameters of $NO_3^-$ uptake by roots of *C. oleifera* seedlings.

| Treatment | Equation | $R^2$ | $Vmax$ $(umol·g^{-1}·h^{-1})$ | $Km$ $(mmol·L^{-1})$ | $\alpha$ |
|---|---|---|---|---|---|
| A0 | $Y = 0.0013x^2 - 0.044x + 2.0386$ | 0.9917 | 16.10 | 1.76 | 0.009 |
| A1 | $Y = 0.0014x^2 - 0.0462x + 2.0373$ | 0.9911 | 17.52 | 1.75 | 0.010 |
| A2 | $Y = 0.0009x^2 - 0.0432x + 2.0290$ | 0.9847 | 16.24 | 1.54 | 0.011 |
| A3 | $Y = 0.0014x^2 - 0.0531x + 2.0414$ | 0.9947 | 17.57 | 1.64 | 0.011 |
| A4 | $Y = 0.0009x^2 - 0.0441x + 2.0424$ | 0.9858 | 16.29 | 1.56 | 0.011 |
| A5 | $Y = 0.0013x^2 - 0.0436x + 2.0525$ | 0.9950 | 15.62 | 1.79 | 0.009 |

The *Km* values of different treatments followed a descending order, with A5 > A0 > A1 > A3 > A4 > A2, i.e., seedlings treated with mixed nitrogen sources (A2–A4) had a lower *Km* compared with seedlings treated with a total-nitrate source (A1), which had a lower *Km* than the control seedlings (A0), which, in turn, had a lower *Km* than total-ammonium-treated seedlings (A5). This indicates that adding a certain proportion of $NH_4^+$ can increase the affinity of *C. oleifera* roots with $NO_3^-$.

The $\alpha$ values of different treatments followed the order A3 = A4 = A2 > A1 > A0 = A5, i.e., mixed-nitrogen-source treatments (A2–A4) produced greater $\alpha$ values in the *C. oleifera* seedlings compared to the total-nitrate treatment (A1), which were greater than those in the control treatment (A0), which, in turn, were greater than the ones resulting from the total-ammonium treatment (A5). The mixed-nitrogen-source treatments (A2–A4) had the largest $\alpha$ value of 0.011, while the single-nitrogen-source treatments (A1, A5) had the smallest $\alpha$ value of 0.009, indicating that the presence of a certain percentage of $NH_4^+$ ions can increase the uptake rate of $NO_3^-$ by the root.

### 3.2.2. Effect of Nitrogen Form on $NH_4^+$ Uptake Kinetics of *C. oleifera* Seedling Roots

Figures 6 and 7 show the curves of $NH_4^+$ concentration at different uptake times in the $NH_4^+$ depletion test of *C. oleifera* seedlings treated with different nitrogen forms. The kinetic response patterns of $NH_4^+$-N uptake by seedlings subjected to different treatments showed no statistical difference. In all samples, the $NH_4^+$ concentration decreased rapidly in the first 8 h, with the most obvious change observed within the first hour. The $NH_4^+$ concentration remained almost unchanged after 8 h. Via fitting, the $NH_4^+$ depletion equations for treatments with different nitrogen forms and ratios were listed in Table 3. The kinetic

parameters of $NH_4^+$ uptake by the roots were calculated according to the fresh weights of the roots of *C. oleifera* seedlings in different treatments.

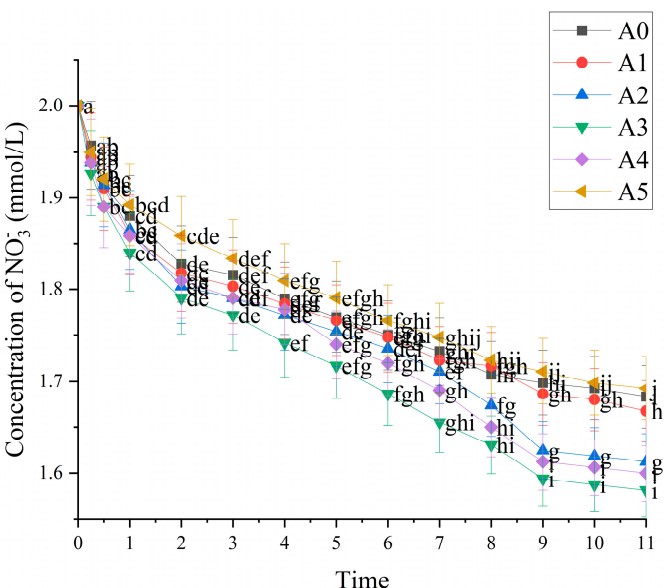

**Figure 6.** Absorption kinetics curve of nitrate in roots of *C. oleifera* seedlings.

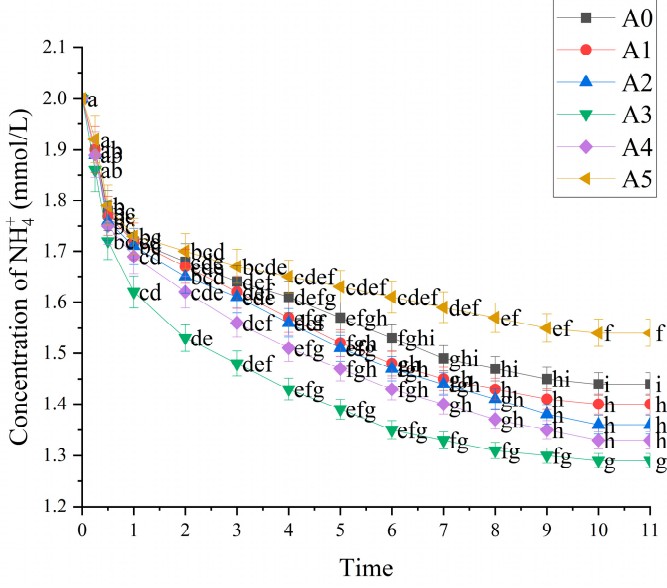

**Figure 7.** Absorption kinetics curve of ammonium in roots of *C. oleifera* seedlings.

**Table 3.** Kinetic parameters of $NH_4^+$ uptake by roots of *C. oleifera* seedlings.

| Treatment | Equation | $R^2$ | Vmax (umol·g$^{-1}$·h$^{-1}$) | Km (mmol·L$^{-1}$) | α |
|---|---|---|---|---|---|
| A0 | $Y = 0.0032 x^2 - 0.0888x + 2.0549$ | 0.9889 | 32.49 | 1.59 | 0.020 |
| A1 | $Y = 0.0037x^2 - 0.0993x + 2.0744$ | 0.9946 | 35.01 | 1.57 | 0.022 |
| A2 | $Y = 0.0034x^2 - 0.0971x + 2.0632$ | 0.9927 | 33.95 | 1.54 | 0.022 |
| A3 | $Y = 0.0057x^2 - 0.1361x + 2.0988$ | 0.9924 | 45.07 | 1.49 | 0.030 |
| A4 | $Y = 0.0041x^2 - 0.1111x + 2.0825$ | 0.9947 | 39.35 | 1.52 | 0.026 |
| A5 | $Y = 0.0032x^2 - 0.0793x + 2.0401$ | 0.9676 | 29.62 | 1.67 | 0.018 |

As shown in Table 3, the *Vmax* values of different treatments followed a descending order, with A3 > A4 > A1 > A2 > A0 > A5. The A3 treatment has the highest *Vmax* of 45.07 $\mu mol \cdot g^{-1} \cdot h^{-1}$, indicating that the A3 treatment had the highest intrinsic potential for $NH_4^+$ uptake. The A3 treatment was followed by the A4 treatment with a Vmax of 39.35 $\mu mol \cdot g^{-1} \cdot h^{-1}$, and then by the A1 treatment. This indicates that the addition of $NO_3^-$ accelerated $NH_4^+$ uptake. As for single-nitrogen-source treatments, the total-nitrate treatment (A1) had a higher *Vmax* than the total-ammonium treatment (A5), i.e., the $NH_4^+$ uptake rate was higher in the total-nitrate treatment.

The *Km* values of different treatments were in the descending order of A5 > A0 > A1 > A2 > A4 > A3, showing that mixed-nitrogen-source treatments (A2–A4) had a lower *Km* value than the total-nitrate treatment (A1), which had a lower *Km* value than control (A0) treatment, which, in turn, had a lower *Km* than the total-ammonium treatment (A5). This indicates that the involvement of a certain percentage of $NO_3^-$ can enhance the affinity of *C. oleifera* roots with $NH_4^+$.

The $\alpha$ values of $NH_4^+$ uptake by the seedlings were ranked as A3 > A4 > A2 = A1 > A0 > A5, with the mixed-nitrogen-sources (A2–A4) and total-nitrate treatments (A1) producing greater $\alpha$ values of $NH_4^+$ uptake compared with control (A0), which had a greater $\alpha$ value of $NH_4^+$ uptake than the total-ammonium treatment (A5). This indicates that involving a certain percentage of $NO_3^-$ can accelerate $NH_4^+$ uptake by the root.

## 4. Discussion

### 4.1. Effects of Nitrogen Form on the Root Activity of C. oleifera Seedlings

Root activity is an important indicator reflecting the nutrient uptake capacity. The total and active absorption area of the root can reflect the strength of root activity and, to some extent, the thickness, branching, and root hair volume of the root [26]. Study of nitrogen form supply on root respiration of walnut seedlings showed that the root respiration rate of walnut seedlings, as well as soluble sugar and starch content, significantly higher than other nitrogen forms treatment when the ratio of ammonium nitrogen to nitrate nitrogen was 50:50 [27]. A study on the effects of exogenous nitrogen forms on cucumber growth showed that the equal amount of ammonium–nitrogen and nitrate–nitrogen promoted the growth of branches, leaves and roots of cucumber seedlings, as well as having enhanced tolerance to sub-low temperatures [28]. In this study, when the ratio of ammonium–nitrogen to nitrate–nitrogen was 50:50, the roots of *C. oleifera* seedlings showed the strongest uptake ability of water and nutrient elements, i.e., the highest root activity, which was consistent with the findings in the other species mentioned above. The root activity was the lowest in the total-ammonium treatment, indicating that the *C. oleifera* root has poor adaptability to a pure $NH_4^+$-N environment. Therefore, mixed nitrogen forms can improve root activity, thus enhancing nitrogen nutrient uptake by the plants.

### 4.2. Effects of Nitrogen Form on $NO_3^-$ and $NH_4^+$ Uptake by Roots of C. oleifera Seedlings

Plants have different uptake and transport pathways for $NH_4^+$-N and $NO_3^-$-N. Extensive studies have shown that high concentrations of $NH_4^+$ inhibit $NO_3^-$ uptake, which may be due to the inhibition of gene expression for $NO_3^-$ carrier protein synthesis, as the presence of $NH_4^+$ affects the environment in which $NO_3^-$ carriers are located on the cell membrane [29]. In this study, low concentrations of $NH_4^+$ accelerated the uptake of $NO_3^-$-N by the roots of *C. oleifera* seedlings. This is likely because a certain concentration of $NH_4^+$ induces $H^+$ secretion by roots, promoting $NO_3^-/H^+$ cotransport, thus accelerating $NO_3^-$ uptake [30]. However, as the $NH_4^+$ concentration gradually increased, the Vmax for $NO_3^-$-N uptake decreased rapidly, which was confirmed by the $NO_3^-$-N uptake characteristics of *C. oleifera* seedlings in this study. This conclusion is consistent with findings in other plants, such as citrange and banana plants [31,32].

The $NO_3^-$-N concentration in the environment can also affect the uptake capacity of plant roots. The presence of $NO_3^-$-N promotes root growth, which, in turn, promotes the uptake and utilization of $NH_4^+$-N by the underground part of the plant [33]. In this study,

the *Vmax* of the total-nitrate treatment (A1) was lower than that of the A3 treatment but higher than all the other treatments, which may be because the root has sufficient $NO_3^-$-N uptake carriers under pure $NO_3^-$ supply [34]. In contrast, the *Vmax* of the total-ammonium treatment (A5) was lower than that of all the other treatments, which is consistent with the findings of Du Xvhua et al. on tea trees and Sun Minhong et al. on citrange [31,35].

The *C. oleifera* seedlings subjected to A3 treatment had higher Vmax and α values for the uptake rates of both $NO_3^-$ and $NH_4^+$ than those in the other treatments, indicating that *C. oleifera* seedlings have a strong preference and competitiveness for both $NH_4^+$-N and $NO_3^-$-N when applied in equal ratios.

By comparing the characteristic parameters of $NH_4^+$-N and $NO_3^-$-N uptake kinetics by *C. oleifera* seedlings treated with different nitrogen forms and ratios, it was found that *Vmax* $NH_4^+$ > *Vmax* $NO_3^-$, i.e., the uptake potential of *C. oleifera* seedlings for $NH_4^+$-N, is greater than that for $NO_3^-$-N. In addition, *Km* $NO_3^-$ > *Km* $NH_4^+$, indicating that *C. oleifera* seedlings have a higher affinity with $NH_4^+$-N than with $NO_3^-$-N. The results of the study show that *C. oleifera* seedlings prefer $NH_4^+$-N, which is likely due to the long-term adaptation of *C. oleifera* to acidic soils. In acidic soils, $NO_3^-$ tends to be rapidly reduced to $NH_4^+$, and $NH_4^+$ becomes the main nitrogen source [36]. However, upon $NH_4^+$ uptake, the plants release $H^+$ into the soil, which will increase the acidity of the soil. Favorable growth of *C. oleifera* seedlings requires soil with a suitable acidity (pH 4.5–6.0), as soil that is too acidic will inhibit seedling growth. During $NO_3^-$ uptake, *C. oleifera* releases $HCO_3^-$ into the soil and raises the pH of the soil. Therefore, applying mixed $NH_4^+$-N and $NO_3^-$-N at an appropriate ratio can stabilize the soil pH and promote the growth of *C. oleifera* seedlings. In addition, $NO_3^-$ is easily mobile and has a high diffusion efficiency, as well as being prone to leaching loss, which reduces the uptake and utilization of $NO_3^-$ by plants.5.

## 5. Conclusions

By comparing the characteristic parameters of different ratios of ammonium– and nitrate–nitrogen with either single ammonium– or nitrate–nitrogen, the mixed nitrogen source was better for promoting the root activity of *C. oleifera* seedlings, while both the total absorption area and active absorption area of the seedling roots were highest with the nitrogen source form of $NO_3^-$:$NH_4^+$ = 5:5. Moreover, *C. oleifera* seedlings have a higher uptake potential and affinity with $NH_4^+$-N than with $NO_3^-$-N, indicating that *C. oleifera* prefers $NH_4^+$-N.

**Author Contributions:** Conceptualization, R.W. and Y.C.; methodology, R.W. and Z.Z.; software, R.W. and T.X.; validation, R.W., X.W. and Y.C.; formal analysis, R.W. and Z.H.; investigation, R.W. and Z.Z.; writing—review and editing, R.W. and C.L.; funding acquisition, R.W. and Y.C. All authors have read and agreed to the published version of the manuscript.

**Funding:** This research was funded by the Major Special Project of Changsha Science and Technology Bureau (KQ2102007), and the Natural Science Foundation of Hunan Province (Grant No. 2022JJ30325 and No. 2021JJ40283).

**Data Availability Statement:** The data presented in this study are available on request from the corresponding author.

**Conflicts of Interest:** The authors declare no conflict of interest.

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
