# Peer review of "Effects of Nitrogen Form on Root Activity and Nitrogen Uptake Kinetics in Camellia oleifera Seedlings"

_forests, doi:10.3390/f14010161_

Round 1
Reviewer 1 Report
The paper entitled “Effects of nitrogen form on root activity and nitrogen uptake kinetics in Camellia oleifera seedlings” investigated the effect of nitrogen on root activity of C. oleifera seedlings and nitrogen uptake kinetics. This study provides a scientific basis for improving nitrogen use efficiency and scientific fertilization in C. oleifera production which is good for local and all the researchers. The paper prepared well but major revision must consider improving it.
Abstract:
1. LN 10: mention the complete scientific name of the species for the first time
2. LN 11: what is Xianglin 27?
3. LN 12: “the effects of nitrogen form were investigated by….” Not clear.
4. LN 22: What is the conclusion?
Introduction:
5. LN 38-40: reference?
6. The introduction is not complete, what is the novelty of study? What are the main and secondary aims? Literature reviews?
Material and methods:
7. Provide a study area map.
8. LN 59: what is that classification “Quaternary red soil”?
9. When did you do the experiment?
10. Table 1: cite and mention table 1 before the table not after it.
11. Table 1: what are the treatments in column 1 “treatment”? you must explain all treatments.
12. Table 1: What is the reference of ratios?
13. Section 2.4.1: it is only one sentence, add this sentence to other sentences. It is not good to be as a section. Maybe it is better to merge three sections in 2.4 as on section.
Results:
14. Figure 1 to figure 5 do not have a good resolution. Replace with a good one.
15. Figure 1 to figure 5: what are the x and y axis title?
16. Figure 6 and figure 7: put the units in a parenthesis.
17. Table 2 and 3: delete the space for R2 values.
18. R2 is not a good parameter to compare treatments. Use RMSE and MAE.
Discussion:
19. The discussion is so poor. Your results are too much but only discussed in two paragraphs. Make the discussion strong compare your results with others and the reasons to be agree of disagree with others results.
Conclusion:
20. Same as discussion, you must write more comprehensive.
Author Response
Dear Professor:
Many thanks for your precious advices helping me improve this manuscript. I have had my manuscript revised follow your suggestions, and listed my answers to you questions below:
1.LN 10: mention the complete scientific name of the species for the first time
It has been corrected in the revision.
2.LN 11: what is Xianglin 27?
‘Xianglin 27’ is one of C. oleifera cultivars that widely cultivated in the middle and south of China.
3.LN 12: “the effects of nitrogen form were investigated by….” Not clear.
It has been corrected in the revision.
4.LN 22: What is the conclusion?
The conclusion was that compare to either ammonium- or nitrate-nitrogen, the mixed nitrogen source was better for promote the root activity of C. oleifera seedlings, and the best nitrate/ammonium ratio was 5:5.
5.LN 38-40: reference?
The reference was added in the revision.
- The introduction is not complete, what is the novelty of study? What are the main and secondary aims? Literature reviews?
The introduction has been rewritten in the revision according to reviewer’s suggestions.
7.Provide a study area map.
A map of the National Oil-tea Camellia Engineering & Technology Research Center was added in the revision.
8.LN 59: what is that classification “Quaternary red soil”?
The Quaternary red soil is widely distributed in the middle and lower reaches of the Yangtze River, and also an important medium reflecting the Quaternary paleo-environment in southern China.
9.When did you do the experiment?
The experiment started in June 2019 with the treatment of fertilization with different nitrogen forms and finished in October 2019.
10.Table 1: cite and mention table 1 before the table not after it.
It has been corrected in the revision.
11.Table 1: what are the treatments in column 1 “treatment”? you must explain all treatments.
Column 1 in Table 1 just reflected that there were six groups of treatment and control experimental materials, it has been deleted in the revision.
12.Table 1: What is the reference of ratios?
The writer of this manuscript has published a study on effects of different proportion of nitrogen forms on the growth and physiological characteristics of Camellia oleifera seedlings in 2019 which has done the experiment back in 2015. The reference has added in the revision.
- Section 2.4.1: it is only one sentence, add this sentence to other sentences. It is not good to be as a section. Maybe it is better to merge three sections in 2.4 as on section.
Section 2.4.1 and 2.4.2 has been merged in the revision.
14.Figure 1 to figure 5 do not have a good resolution. Replace with a good one.
It has been replaced in the revision.
15.Figure 1 to figure 5: what are the x and y axis title?
It has been replaced in the revision.
16.Figure 6 and figure 7: put the units in a parenthesis.
It has been replaced in the revision.
17.Table 2 and 3: delete the space for R2 values.
It has been collected in the revision.
- R2 is not a good parameter to compare treatments. Use RMSE and MAE.
The equations of Table2 and 3 were calculated to evaluate Km, Vmax and α, which reflected the affiliations, absorption properties and rate of nitrogen uptake in root system, respectively. R2 is a parameter that evaluate the accuracy of these equations.
19.The discussion is so poor. Your results are too much but only discussed in two paragraphs. Make the discussion strong compare your results with others and the reasons to be agree or disagree with others results.
The discussion section was rewritten in the revision.
20.Same as discussion, you must write more comprehensive.
The conclusion section was rewritten in the revision.
Reviewer 2 Report
Dear authors,
In this paper, the authors investigate the effect of different nitrate/ammonium ratios as fertilizers on different parameters of Camellia oleifera. Their results show that a certain nitrate/ammonium mixture gives the best results. The paper is generally quite well written. However, I consider that the authors need to improve certain aspects that I indicate as essential.
Majors:
Introduccion:
It should be substantially improved. I think that the introduction gives too little information, for example, some data that I think are missing. Is the genome of this species sequenced? Is anything known about the genes that transport nitrate or ammonium, have they been identified? What is known about the genes for ammonium or nitrate transporters in related organisms? What is the economic importance of Camellia oleifera in that region or in others?
Materials and methods: It is essential to indicate which methods were used to determine: total absorption area, the active absorption area, active absorption area ratio, specific surface area, active specific surface area
L96-L102: “Determination of kinetic characteristics of NO-3-N and NH+4 -N uptake” At what stage of plant development was the experiment performed? please indicate
Figure 3, Statistical study of these data is lacking
Section 3.2: Why only nitrate and ammonium transport measurements were carried out for 11 hours? Why were no samples taken after 24 hours or the following days?
Figure 6, Figure 7: Need to show error bars and statistical treatment.
L144: “on the specific Surface” should be change to: “active absorption area ratio”
Discussion:
L282-L291: This paragraph discusses the results in a very general and poor way. Please, it is fundamental to know the importance of such data, it is necessary to make a broader reference of the data obtained with the available bibliography.
Section 3.2: Why was 2 mM of nitrate or ammonium chosen to carry out the transport experiments? Why not 20 or 0,2 mM, What is this choice based on? Please clarify? Why not use a lower concentration? as the data shows the transport is saturated between 1.2-1.5 mM nitrate or ammonium. Why weren't 2 or 3 different high and low ammonium or nitrate concentrations used to study whether there are different kinetics?
L270: “3.3. Nitrogen uptake characteristics of C. oleifera seedlings” I don't see why this is shown as a separate paragraph? since what is analyzed are the previous results. Please put it better as one more part of the discussion
Minors:
Table 1, column Total nitrogen/mmol·L-1, only those of condition 1 and 4 are indicated, please check
L82: Please, leave space with the table above, it looks like a table footer
L95: “to determine root activity” what kind of activity? clarify it please
L118: “(A2-A4, [m(NO3-N)/m(NH+ 4 -N)]=7:3, 5:5” A few words above you have indicated that for simplicity you would only use A0 and 1-5, please be consistent
Figure 1, Figure 2, Figure 3, Figure 4, the legend of the Y axis must be completed, m2 of what?? please fix. It is better to indicate under each column that they are, A0....5, in this way the legend of the color of the columns will not be necessary, remove it. please change.
Fig 4, Fig 5: what does this mean cm2 cm-3?? please specify
L205-L208, L243-L245: I don't see the importance of indicating these equations here, what is it? they are already shown in table 2 and 3, please remove them
Table 2, please indicate what is α also in the table
L296: “NH+ 4” There are some typographical errors like this that I give as an example, please check all of them
Author Response
Dear Professor:
Many thanks for your precious advices helping me improve this manuscript. I have had my manuscript revised follow your suggestions, and listed my answers to you questions below:
- It should be substantially improved. I think that the introduction gives too little information, for example, some data that I think are missing. Is the genome of this species sequenced? Is anything known about the genes that transport nitrate or ammonium, have they been identified? What is known about the genes for ammonium or nitrate transporters in related organisms?
The introduction has been rewritten in the revision according to reviewer’s suggestions.
- What is the economic importance of Camellia oleifera in that region or in others?
The introduction has been rewritten in the revision according to reviewer’s suggestions.
- Materials and methods: It is essential to indicate which methods were used to determine: total absorption area, the active absorption area, active absorption area ratio, specific surface area, active specific surface area
It has been added in the revision
- L96-L102: “Determination of kinetic characteristics of NO-3-N and NH+4 -N uptake” At what stage of plant development was the experiment performed? please indicate
The kinetic characteristics of the uptake of NO- 3-N and NH+ 4-N solutions by seedlings were determined by the conventional depletion method during the late of July to the middle of august when was the most active stage of seedlings' growth.
- Figure 3, Statistical study of these data is lacking
It has been replaced in the revision.
- Section 3.2: Why only nitrate and ammonium transport measurements were carried out for 11 hours? Why were no samples taken after 24 hours or the following days?
In this study, the concentration of NO- 3 and NH+ 4 hardly changed since 9 and 8 hours after treatment, respectively, so we stopped sampling at 11 hours after treatment.
- Figure 6, Figure 7: Need to show error bars and statistical treatment.
It has been replaced in the revision.
- L144: “on the specific Surface” should be change to: “active absorption area ratio”
It has been corrected in the revision.
- L282-L291: This paragraph discusses the results in a very general and poor way. Please, it is fundamental to know the importance of such data, it is necessary to make a broader reference of the data obtained with the available bibliography.
This part was rewritten in the revision.
- Section 3.2: Why was 2 mM of nitrate or ammonium chosen to carry out the transport experiments? Why not 20 or 0.2 mM, What is this choice based on? Please clarify? Why not use a lower concentration? as the data shows the transport is saturated between 1.2-1.5 mM nitrate or ammonium. Why weren't 2 or 3 different high and low ammonium or nitrate concentrations used to study whether there are different kinetics?
Some references were added to this manuscript in the revision to support we chose 2 mM as a suitable condition in this study.
- L270: “3.3. Nitrogen uptake characteristics of C. oleifera seedlings” I don't see why this is shown as a separate paragraph? since what is analyzed are the previous results. Please put it better as one more part of the discussion
This part has been put into discussion section.
- Table 1, column Total nitrogen/mmol·L-1, only those of condition 1 and 4 are indicated, please check
It has been adjusted in the revision.
- L82: Please, leave space with the table above, it looks like a table footer
It has been adjusted in the revision.
- L95: “to determine root activity” what kind of activity? clarify it please
The Methylene Blue method was generally applied to determine the absorption activity of root system. It was clarified in the revision.
- L118: “(A2-A4, [m(NO3-N)/m(NH+ 4 -N)]=7:3, 5:5” A few words above you have indicated that for simplicity you would only use A0 and 1-5, please be consistent
It has been collected in the revision.
- Figure 1, Figure 2, Figure 3, Figure 4, the legend of the Y axis must be completed, m2 of what?? please fix. It is better to indicate under each column that they are, A0. 5, in this way the legend of the color of the columns will not be necessary, remove it. please change. Fig 4, Fig 5: what does this mean cm2 cm-3?? please specify
These figures have been replaced in the revision.
- L205-L208, L243-L245: I don't see the importance of indicating these equations here, what is it? they are already shown in table 2 and 3, please remove them
It has been collected in the revision.
- Table 2, please indicate what is α also in the table
The equations of Table2 and 3 were calculated to evaluate Km, Vmax and α, which reflected the affiliations, absorption properties and rate of nitrogen uptake in root system, respectively.
- L296: “NH+ 4” There are some typographical errors like this that I give as an example, please check all of them
It has been collected in the revision.
Round 2
Reviewer 1 Report
Good job!
Reviewer 2 Report
The authors have responded correctly to all my suggestions and I accept the paper in its latest version.